# *PBX3* as a biomarker for the early diagnosis and prediction of prognosis of glioma

**Cuicui pan**[1], **Xueli bai**[1], **Na Li**[2], **Ni Zheng**[1], **Yuanquan Si**[1]*, **Yueran Zhao**[3]*

**1** Department of Clinical Laboratory, Shandong Provincial Hospital Affiliated to Shandong First Medical University, Jinan, China, **2** Department of Dermatology, The Affiliated Hospital of Shandong University of Traditional Chinese Medicine, Jinan, China, **3** Central Laboratory, Shandong Provincial Hospital affiliated to Shandong University, Jinan, China

* siyuanquan123@163.com (YS); yrzhao@sdu.edu.cn (YZ)

**Data Availability Statement:** All relevant data are within the paper and its Supporting information files.

**Funding:** There are no financial conflicts of interest to disclose.

## Abstract

### Background

Increasing evidence have elucidated that *PBX3* played a crucial role in cancer initiation and progression. *PBX3* was differentially expressed in many cancer types. However, PBX3 potential involvement in gliomas remains to be explored.

### Methods

The expression level of *PBX3* in glioma tissues and glioma cells, and its correlation with clinical features were analyzed by data from TCGA, GEPIA, CGGA and CCLE. Univariable survival and Multivariate Cox analysis was used to compare several clinical characteristics with survival. We also analyzed the correlation between *PBX3* expression level and survival outcome and survival time of LGG and GBM patients by using linear regression equation. GSEA was used to generate an ordered list of all genes related to *PBX3* expression and screening of genes co-expressed with PBX3 mRNA by "limma" package.

### Results

The results showed that *PBX3* was highly expressed in gliomas and its expression increased with the increase of malignancy. Survival analysis found that *PBX3* is more valuable in predicting the OS and PFI of LGG patients than that of GBM. For further study, TCGA and CGGA data were downloaded for univariate Cox analysis and multivariate Cox analysis which showed that the expression of *PBX3* was independent influencing factors for poor prognosis of LGG patients. Meanwhile, Receiver operating characteristic (ROC) curve showed that *PBX3* was a predictor of overall survival rate and progression-free survival rate of LGG. Linear regression model analysis indicated that the higher expression of *PBX3* the higher the risk of death of LGG patients, and the higher expression of PBX3 the higher the risk of disease progression of LGG patients. Next, TCGA data were downloaded for GSEA and Co-expression analyses, which was performed to study the function of *PBX3*.

**Competing interests:** The authors have declared that no competing interests exist.

**Abbreviations:** CCLE, Cancer Cell Line Encyclopedia; CGGA, Chinese Glioma Genome Atlas; GBM, Glioblastoma; GEPIA, Gene Expression Profiling Interactive Analysis; GSEA, Gene Set Enrichment Analysis; GTEx, Genotype-Tissue Expression project; IDH, Isocitrate Dehydrogenase; LGG, Low-Grade Glioma; RSEM, RNA-Seq by Expectation Maximization; TCGA, The Cancer Genome Atlas; UCSC, University California santa cruz.

## Conclusion

*PBX3* may be involved in the occurrence and development of glioma, and has potential reference value for the early diagnosis and prediction of prognosis of glioma.

## Introduction

Glioma which originates from glial cells is the most common malignant tumor in the human central nervous system (CNS), accounting for about 29% of all brain tumors and 80% of intracranial malignant tumors [1]. The median survival time of glioblastoma is less than 1.5 years, and the 5-year survival rate is only 0.05%-4.7% [2, 3]. At present, the diagnosis of gliomas is mainly based on clinical manifestations, imaging examination and pathological analysis. The treatment of glioma mainly includes surgical resection combined with postoperative radiotherapy and chemotherapy. The glioma of G1 and G2 grows slowly, have a good prognosis, and are generally considered to be benign. They can be completely respected and even completely recovered from surgery. The glioma of G3 and G4 which have the highest incidence rate grow rapidly and spread easily to other brain tissues. As a result, it is easy to relapse and has the poor prognosis. Thus, it is quite urgent to investigate the mechanisms underlying the development and progression of glioma in order to identify sensitive and specific early biomarkers for diagnosis and prognosis.

Pre-B cell leukemia homeobox 3 (*PBX3*) is a member of the PBX family, which can be bind to (*HOX*) proteins, enhance the DNA binding / transcriptional activity of *HOX* proteins, and regulate the transcription and expression of target genes. It is reported that *PBX3* is highly expressed in a variety of tumors, such as prostate cancer, colorectal cancer, gastric cancer, hepatocellular carcinoma and multiple myeloma. Moreover, the expression level of *PBX3* was related to tumor grade and pathological stage [4–6]. Functional studies have shown that the expression of *PBX3* is related to the malignant biological phenotype of tumors. For example, high expression of *PBX3* can promote tumor cell proliferation [7–9], invasion and migration [4, 9], stem cell characteristics [10]. Xu et al. reported that *PBX3* is highly expressed in GBM and promotes GBM proliferation, invasion, migration and apoptosis [6, 11]. In our previous studies, we found that *PBX3* is highly expressed in gliomas, and promote glioma cell proliferation and inhibits apoptosis. However, the expression level of *PBX3* correlation with clinical features, and the correlation between *PBX3* and the prognosis of gliomas has not been studied.

Herein, the relationship between the expression level of *PBX3* and glioma was investigated from the perspective of bioinformatics. First, we analyzed the expression of *PBX3* in glioma tissue, the relationship between *PBX3* and glioma patient's clinical characteristics, and the prognostics value of *PBX3* by TCGA, CGGA and GEPIA database. Then, TCGA and GTEx data were used to comprehensively analyze the clinical correlations. TCGA data were downloaded for differently expressed genes (DEGs) identification, GSEA and Co-expression analyses was performed to study the function of genes positively related to expression of *PBX3*.

## Materials and methods

### Survival and expression analysis by GEPIA

Gene Expression Profiling Interactive Analysis (GEPIA) (http://gepia.cancer-pku.cn/index.html), an online database was used to confirm the correlation between *PBX3* expression and clinicopathologic information in glioma. GEPIA uses a standard processing pipeline that analyzes 9,736 tumors from the RNA sequencing expression data and 8,587 normal samples from

projects known as the Genotype-Tissue Expression project (GTEx) and The Cancer Genome Atlas (TCGA). Box Plots using disease state (Tumor or Normal) as variable was graphed to calculate differential expression of *PBX3*. Meanwhile, Survival curve of differential *PBX3* expression were analyzed to find the correlation of the gene expression with glioma patients' prognosis.

## Expression level of *PBX3* in normal tissues by GTEx data

GTEx were downloaded from University California santa cruz (UCSC) Xena project (http://xena.ucsc.edu/). The data include TOIL RNA-Seq by Expectation Maximization (RSEM) TPM (n = 7,862) and GTEx phenotype (n = 9,783). TCGA data were downloaded from UCSC Xena project. The data include TOIL RSEM TPM (n = 10,535) and Curated clinical data (n = 12,591). The Ensembl number of the gene in RNAseq data was transformed into the Official Symbol of the gene by R 4.2.0. In addition, the GTEx and TCGA data was used to analyze the differential expression of *PBX3* in normal brain tissue and glioma tissue through ggpubr package.

## Clinical correlation analyses between *PBX3* expression and glioma patients from TCGA database

The relevant clinical information of 1111 glioma patients (LGG+GBM) (Low-Grade Glioma and Glioblastoma), was download from TCGA database by RTCGA. clinical package. Meanwhile, we obtained 1131 clinical information of glioma patients from Jianfang's study [12]. Then, after combining the two sets of data, a total of 1131 patients with gliomas were included in this study. The expression value of *PBX3* in glioma patients with clinical information were integrated to obtain 693 patients via perl script (Table 1), and the expression difference of *PBX3* in discrete variables of clinical information was analyzed by using R 4.2.0. Finally, among these 693 patients, we eliminated some patients with incomplete clinical information and obtained 452 patients for COX analysis.

## Clinical correlation analyses between *PBX3* expression and glioma patients from CGGA database

The relevant clinical information of 1019 glioma patients (LGG+GBM) were download from Chinese Glioma Genome Atlas (CGGA) database. The expression value of *PBX3* in glioma patients with clinical information were integrated to obtain 929 patients via perl script (Table 2), and the expression difference of *PBX3* in discrete variables of clinical information was analyzed by using R 4.2.0. Finally, among these 929 patients, we eliminated some patients with incomplete clinical information and obtained 499 patients for COX analysis.

## Expression of *PBX3* in glioma cell lines

The gene expression data of 57 glioma cell lines were downloaded from Cancer Cell Line Encyclopedia (CCLE)databases(https://sites.broadinstitute.org/ccle). The expression difference of *PBX3* in glioma cells was analyzed by using R 4.2.0 ggplot2 packages.

## Linear regression analysis

Based on the data from TCGA and CGGA databases, we analyzed the correlation between *PBX3* expression level and survival outcome and survival time of LGG by using linear regression equation. The analysis was performed with the statistical software packages R and EmpowerStats (http://www.empowerstats.com, X&Y Solutions, Inc., Boston, MA).

**Table 1. Summary of patient cohort information in TCGA.**

| Variables | Case | Percentage |
|---|---|---|
| Primary disease | | |
| LGG | 527 | 76.05% |
| GBM | 166 | 23.95% |
| histological_type | | |
| Astrocytoma | 196 | 28.32% |
| oligoastrocytoma | 133 | 19.22% |
| oligodendroglioma | 198 | 28.62% |
| GBM | 165 | 23.84% |
| Grade | | |
| G2 | 258 | 49.05% |
| G3 | 268 | 50.95% |
| Sample type | | |
| cancer | 666 | 96.1% |
| recurrent | 27 | 3.9% |
| Idh1_mutation | | |
| no | 34 | 27.2% |
| yes | 91 | 72.8% |
| Age | | |
| <55 | 471 | 67.97% |
| ≧55 | 222 | 32.03% |
| Gender | | |
| Female | 295 | 42.57% |
| Male | 398 | 57.43% |
| Cancer status | | |
| tumor free | 197 | 31.98% |
| with tumor | 419 | 68.02% |
| Karnofsky_performance_score | | |
| ≦60 | 51 | 11.92% |
| >60 | 377 | 88.08% |

## GSEA enrichment analysis

Gene Set Enrichment Analysis (GSEA) is a computational method to determine whether a group of genes are differentially expressed in two biological states. In this study, GSEA was used to generate an ordered list of all genes related to *PBX3* expression, and then to identify the differential expression of signal pathways between gliomas with high *PBX3* expression and low *PBX3* expression in TCGA database. Gene set permutations were performed 1000 times for each analysis. The expression level of *PBX3* was used as a phenotype label. The nominal P value and normalized enrichment score (NES) were used to sort the pathways enriched in each phenotype.

## Analysis of gene co-expression of *PBX3* mRNA

Screening of genes co-expressed with *PBX3* mRNA by "limma" package in R4.2.0 software. The threshold of co-expression was correlation coefficient > 0.5 and P < 0.001. Meanwhile, draw a single gene correlation map. The first five genes with positive and negative correlation with *PBX3*mRNA expression were screened, and the correlation circle diagram was drawn.

**Table 2. Summary of patient cohort information in CCGA.**

| Variables | Case | Percentage |
|---|---|---|
| Disease | | |
| LGG | 579 | 62.33% |
| GBM | 350 | 37.67% |
| histological_type | | |
| A | 93 | 10.05% |
| AA | 88 | 95.14% |
| AO | 55 | 59.46% |
| AOA | 162 | 17.51% |
| O | 53 | 57.30% |
| OA | 124 | 13.41% |
| GBM | 350 | 37.84% |
| Grade | | |
| WHO II | 270 | 29.19% |
| WHO III | 305 | 32.94% |
| WHO IV | 350 | 37.84% |
| PRS type | | |
| Primary | 620 | 67.03% |
| Recurrent | 275 | 29.73% |
| Secondary | 30 | 3.24%- |
| IDH_mutation | | |
| Mutant | 484 | 54.75% |
| Wildtype | 400 | 45.25% |
| Age | | |
| <55 | 744 | 80.17% |
| ≧55 | 184 | 19.83% |
| Gender | | |
| Female | 380 | 40.9% |
| Male | 549 | 59.1% |
| 1p19q_codeletion | | |
| Codel | 191 | 31.31% |
| Non-codel | 419 | 68.69% |

## Results

### The expression of *PBX3* in glioma tissues, glioma cells and normal tissues

In order to determine the expression level of *PBX3* in normal human brain tissues, GTEx data analysis was conducted. As shown in Fig 1A–1C, the expression of *PBX3* was observably upregulated in full-grade gliomas, LGG and GBM than that in normal tissues. According to GEPIA database, we found that the expression of *PBX3* was observably upregulated in LGG and GBM than that in normal control (Fig 1D). The expression of *PBX3* was observably upregulated in GBM than that in LGG according to TCGA and CGGA databases (Fig 1E and 1F). For further study, we found that the expression of *PBX3* was related to histological types. There were GBM, astrocytoma, oligoastrocytoma and oligodendroglioma histological types from TCGA databases, the expression of *PBX3* was highest in GBM, followed by astrocytoma, and the expression of *PBX3* in oligoastrocytoma and oligodendroglioma was significantly lower than that of GBM and astrocytoma (Fig 1G). In addition, there were GBM(glioblastoma), A

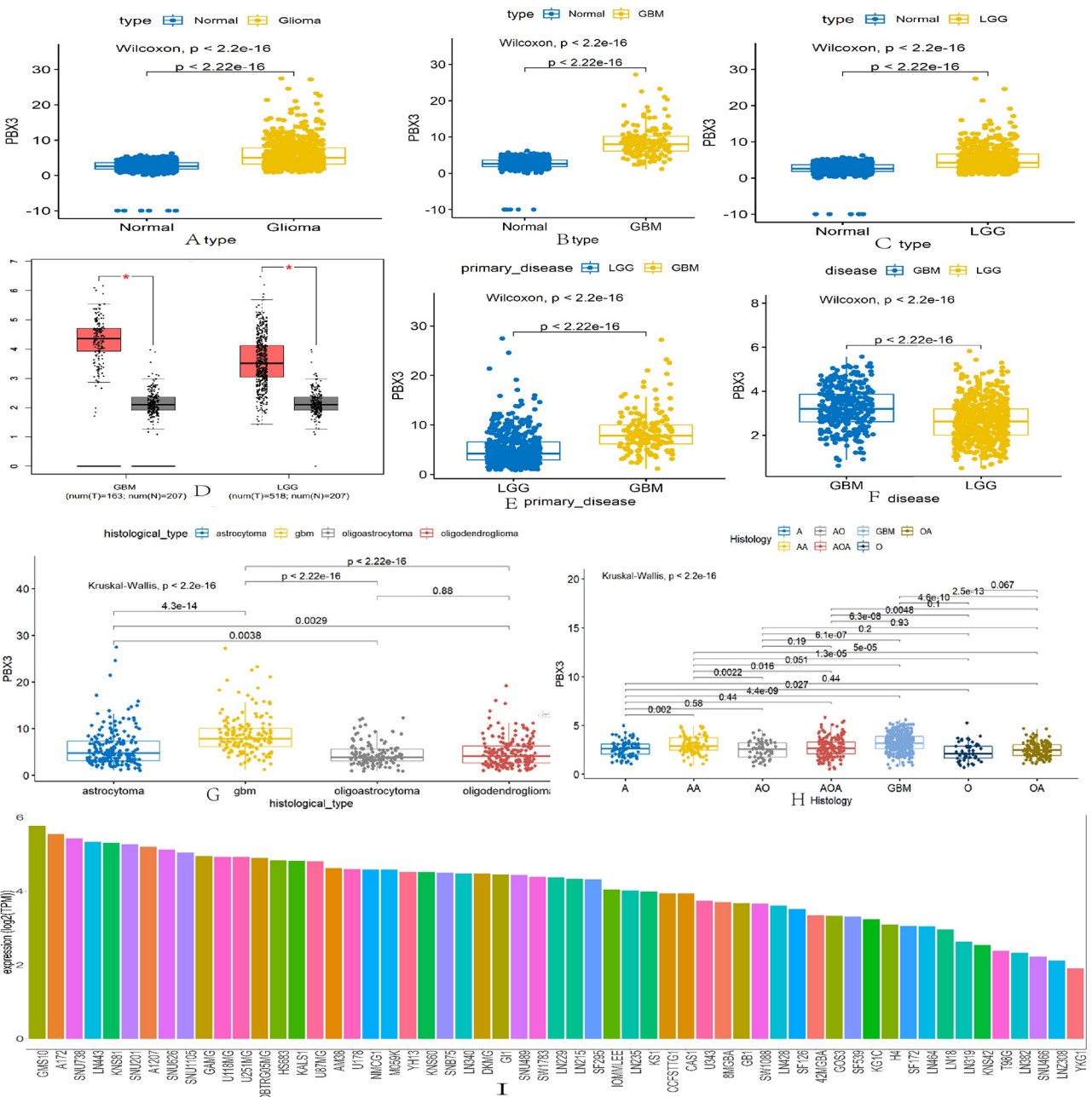

**Fig 1. Expression level of *PBX3* in normal brain, glioma tissues and glioma cells by analyzing data from TCGA, GTEx, GEPIA and CCLE databases.** (A-C) Expression level of *PBX3* in full-grade gliomas, LGG and GBM in comparison with the normal brain tissues according to TCGA and GTEx. (D) Expression level of *PBX3* in LGG and GBM in comparison with the normal brain tissues according to GEPIA. (E) and (F) The expression level of *PBX3* between LGG and GBM according to TCGA and CGGA databases. (G) and (H) The expression level of *PBX3* in histological types of gliomas according to TCGA and CGGA databases. (I) The expression level of *PBX3* in glioma cell lines according to CCLE databases.

(astrocytoma),O(oligodendroglioma),OA(oligo-astrocytoma), AOA(anaplastic oligo-astrocytoma),AA(anaplastic astrocytoma) histological types from CGGA databases. The expression of *PBX3* was highest in GBM, followed by AA, and the expression of *PBX3* in O and OA was significantly lower than that other five histological types, which was consistent with TCGA databases (Fig 1H). By analyzing the expression level of *PBX3* in 57 glioma cell lines from CCLE

database, it was found that the expression of *PBX3* was increased in glioma cell lines (Fig 1I). *PBX3* might be an important molecular mechanism for the occurrence of gliomas.

## Relationship of *PBX3* expression with clinical parameters of patients with glioma

Based on TCGA and CGGA databases, the expression of *PBX3* increased with the increase of glioma histologic-grade (Fig 2A and 2B). The expression of *PBX3* in recurrent glioma patients was significantly higher than that in primary glioma patients, and the expression of *PBX3* was increased with the increase of recurrence times (Fig 2C and 2D). The expression of *PBX3* was higher in IDH wildtype glioma patients than Isocitrate Dehydrogenase (IDH) mutant (Fig 2E and 2F), and the prognosis of IDH wildtype is worse than IDH mutant type. The expression level of *PBX3* in glioma patients age≥55 years was significantly higher than that in patients age <55 years (Fig 2G and 2H). In addition, the expression level of *PBX3* in glioma patients with tumor was higher than that in patients without tumor and the expression level of *PBX3* in glioma patients ((Fig 2I) and karnofsky performance score ≦60 was significantly higher than that in patients karnofsky performance score >60 according to TCGA(Fig 2J). Studies have confirmed that glioma patients with 1p19q-codel have a better prognosis than that non-codeled.

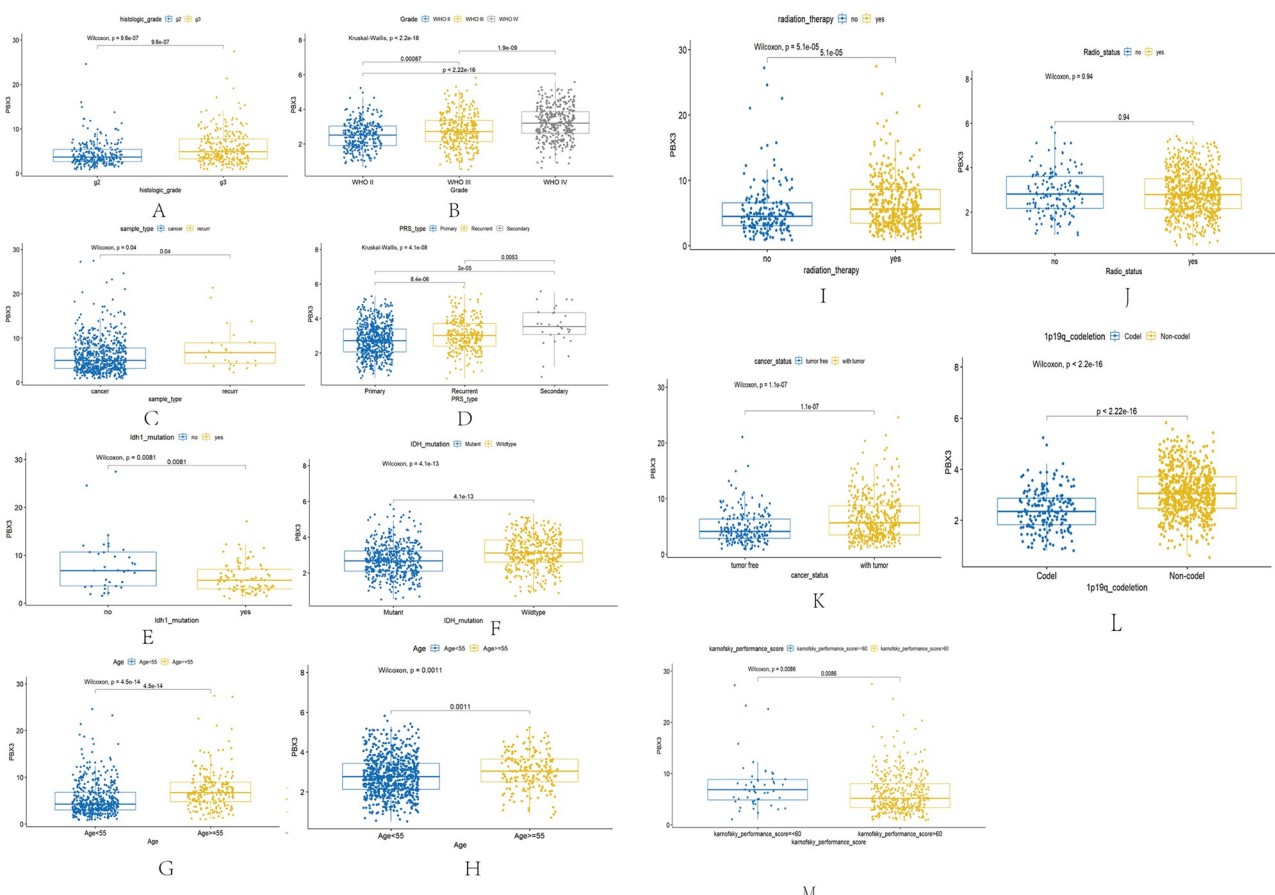

**Fig 2. Correlation analysis between *PBX3* differential expression and clinical features according to TCGA and CGGA.** A, histologic-grade by TCGA; B, histologic-grade by CGGA; C, sample type by TCGA; D, sample type by CGGA; E, IDH-mutant by TCGA; F, IDH-mutant by CGGA; G, Age by TCGA; H, Age by CGGA; (I and J), cancer-status and karnofsky performance score by TCGA; K, 1p19q-codeletion by CGGA.

At the same time, it is found that the expression level of *PBX3* in glioma patients with 1p19q-codel deletion is significantly lower than that non-codel according to CGGA database (Fig 2K). This shows that *PBX3* is associated with tumor progression and has a certain prognostic value in patients with glioma.

## Relationship between *PBX3* expression and overall survival time of patients with glioma

In TCGA and CGGA database, the OS (Overall survival time) of *PBX3* high expression group was shorter than that of low expression group in full-grade gliomas (Fig 3A and 3B). And, in TCGA, CGGA and GEPIA database, the OS of *PBX3* high expression group was shorter than that of low expression group in LGG patients (Fig 3C–3E). In addition, the OS of GBM in low expression of *PBX3* was better than that in high expression of *PBX3* according to CGGA, but there was no statistically significant from GEPIA and TGGA analysis (Fig 3F–3H). This shows that *PBX3* is more valuable in predicting the OS of LGG patients than that of GBM.

## Relationship between *PBX3* expression and PFI (Progression-free interval) of patients with glioma

In TCGA database, the PFI (Progression-free interval) of *PBX3* high expression group was shorter than that of low expression group in full-grade gliomas (Fig 4A). And, in TCGA and GEPIA database, the PFI of *PBX3* high expression group was shorter than that of low expression group in LGG patients (Fig 4B and 4C). In addition, the PFI of GBM in low expression of *PBX3* was better than that in high expression of *PBX3* according to GEPIA, but there was no statistically significant from TGGA analysis (Fig 4D and 4E). This shows that *PBX3* is more valuable in predicting the PFI of LGG patients than that of GBM.

## ROC curve survival analysis of *PBX3*

According to TCGA database, ROC curve analysis showed that the AUC of *PBX3* mRNA predicting 1-year, 3-year and 5-year overall survival rate of LGG was 0.811, 0.791 and 0.698(Fig 5A). And, in CGGA database, ROC curve analysis showed that the AUC of *PBX3* mRNA predicting 1-year, 3-year and 5-year overall survival rate of LGG was 0.695,0.715 and 0.726(Fig 5B). In addition, ROC curve analysis also showed that the AUC of *PBX3* mRNA predicting 1-year, 3-year and 5-year progression-free survival rate of LGG was 0.732,0.693 and 0.658(Fig 5C). This result from TCGA and CGGA database confirmed that the expression of *PBX3* was a predictor of overall survival rate and progression-free survival rate of LGG.

## Univariate and multivariate cox regression analysis of OS prognostic factors in LGG

To further confirm our conjecture, univariate cox analysis was employed in TCGA database. The analysis found that *PBX3* is a highly risk factor (HR = 1.15) for predicting the OS of LGG patients. Besides, age, histological type, caner status, histologic grade were all high-risk factors (Table 3). Then multivariate analysis was performed, and it was unearthed that among these factors, *PBX3*(HR = 1.14) remained independently related to OS, suggesting that *PBX3* might be an independent OS prognostic factor for LGG patients (Table 3).

And, we further confirmed above result by univariate and multivariate analysis according to CGGA. Univariate analysis found that *PBX3* is a highly risk factor (HR = 2.43) for predicting the OS of LGG patients. Besides, age, histological type, caner status, histologic grade were all high-risk factors. Besides, PRS-type (primary, recurrent), IDH-mutation (mutant,

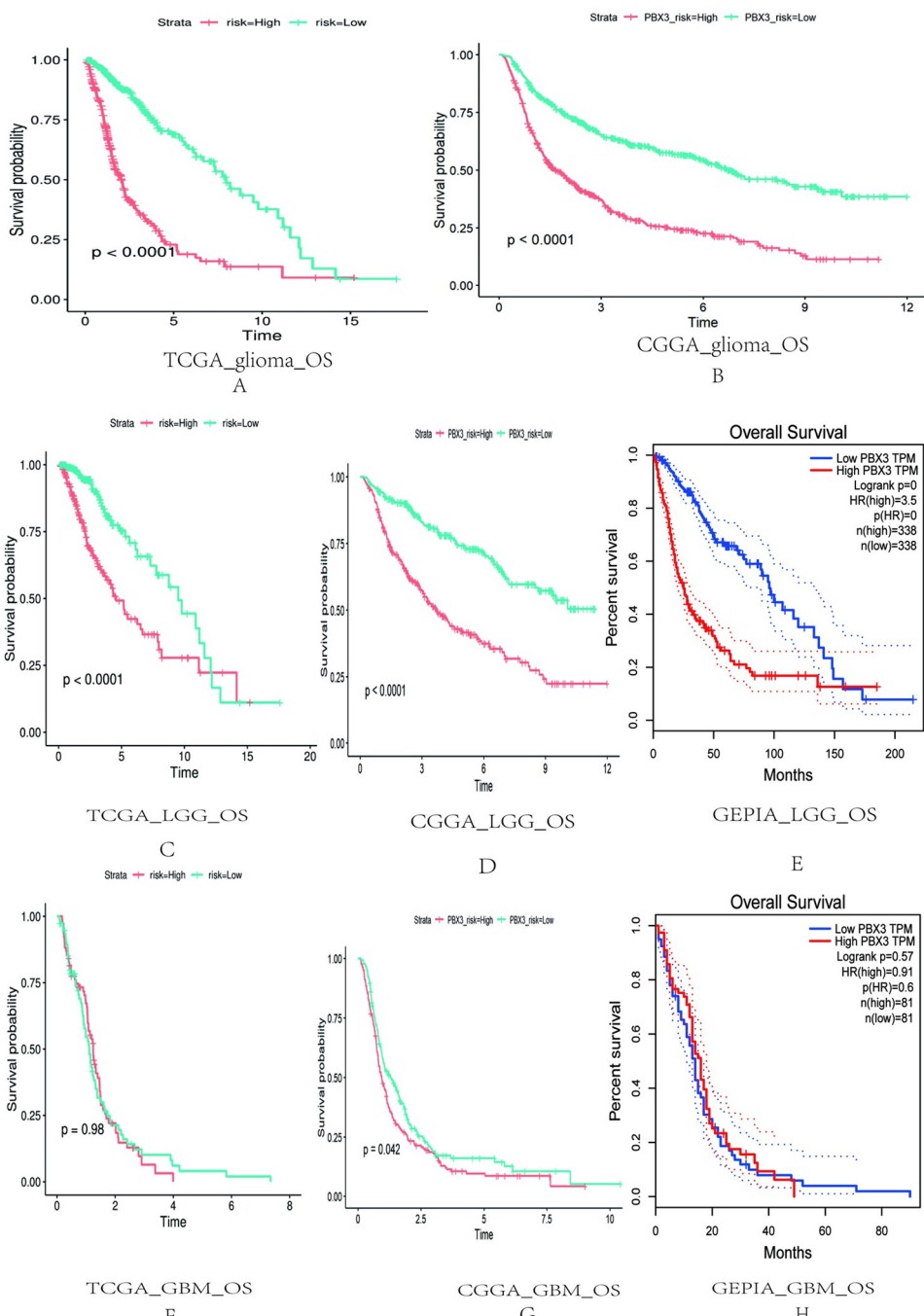

**Fig 3. Relationship between *PBX3* expression and OS of patients with glioma.** (A and B), full-grade gliomas according to TCGA and CGGA databases;(C-E), LGG according to TCGA, CGGA and GEPIA databases;(F-H), GBM according to TCGA, CGGA and GEPIA databases.

wildtype), histological type, WHO grade, 1p19q-codeletion status and age (Table 4). And subsequent multivariate analysis shows that *PBX3* (HR = 1.95) remained independently related to OS, which also suggested that *PBX3* might be an independent OS prognostic factor for LGG patients (Table 4).

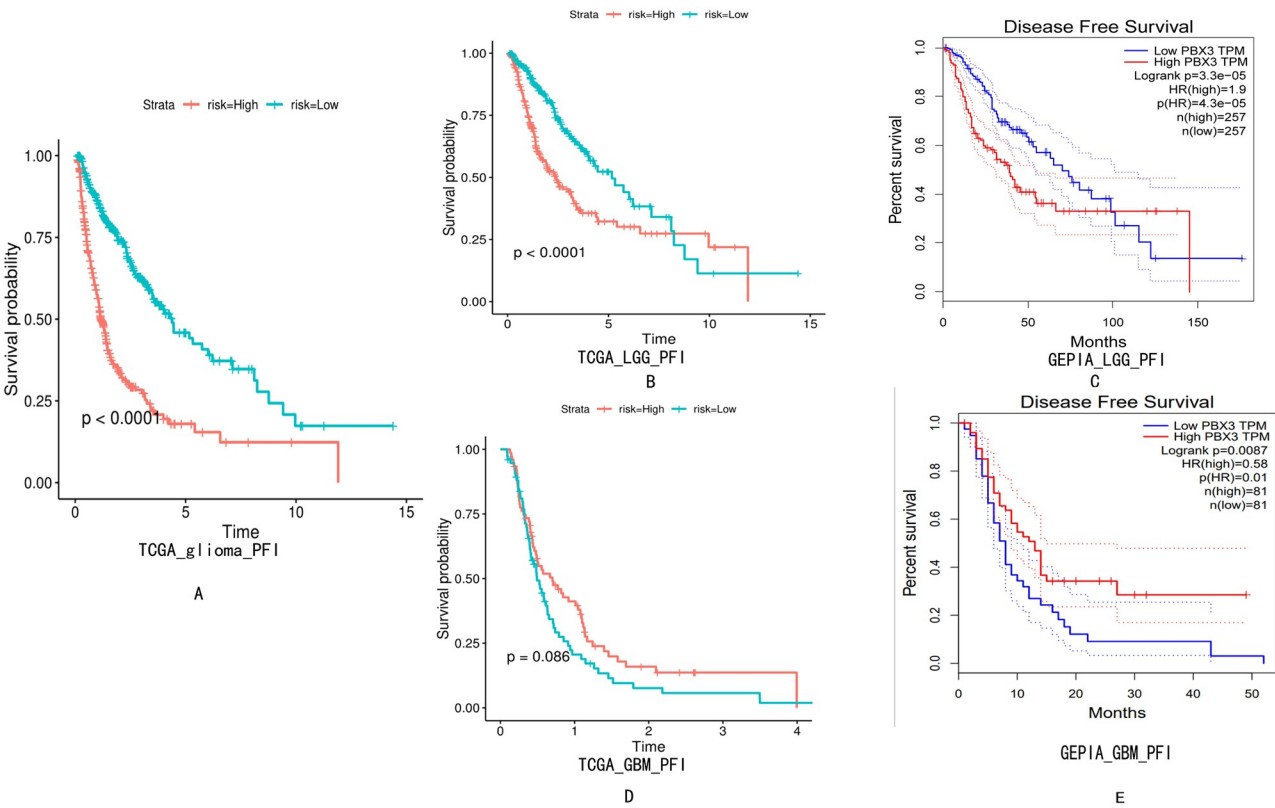

**Fig 4. Relationship between *PBX3* expression and PFI of patients with glioma.** (A), full-grade gliomas according to TCGA databases;(B and C), LGG according to TCGA and GEPIA databases;(D and E), GBM according to TCGA and GEPIA databases.

## Univariate and multivariate cox regression analysis of PFI prognostic factors in LGG

According to TCGA database, univariate analysis found that *PBX3* is a highly risk factor (HR = 1.15) for predicting the PFI of LGG patients. Besides, age, histological type, caner status,

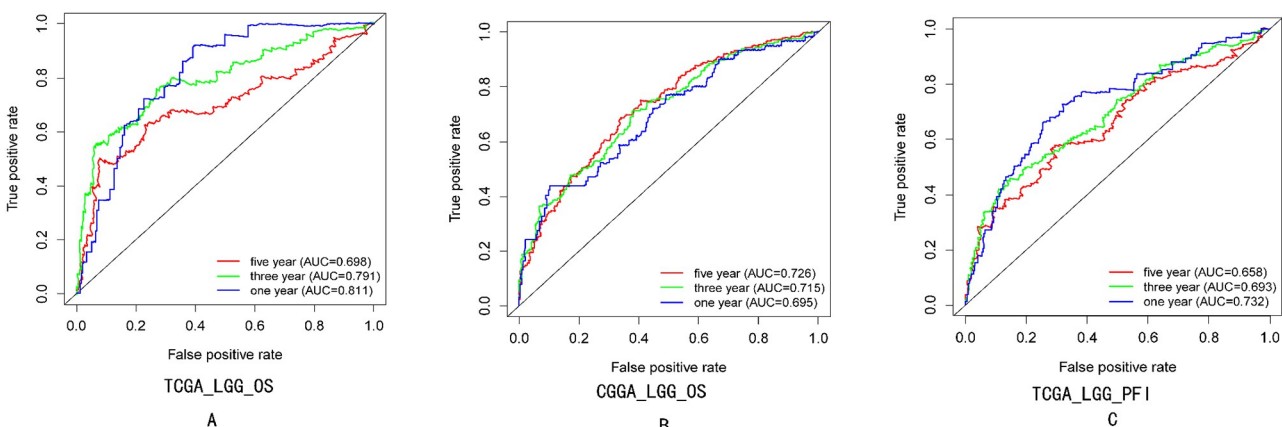

**Fig 5. ROC curve analysis of the predictive efficacy of *PBX3* mRNA on 1 -, 3-and 5-year overall survival rate and progression-free survival rate of LGG.** (A)and(B), overall survival rate;(C) progression-free survival rate.

**Table 3. Univariate analysis and multivariate analysis of OS prognostic factors in LGG according to TCGA.**

| Parameter | Univariate analysis | | | Multivariate analysis | | |
|---|---|---|---|---|---|---|
| | HR | 95% CI | P | HR | 95% CI | P |
| *PBX3* | 1.15 | 1.11–1.20 | 1.83E-15 | 1.14 | 1.09–1.18 | 3.06E-10 |
| Age | 4.19 | 2.84–6.19 | 6.08E-13 | 4.62 | 3.05–7.00 | 5.22E-13 |
| Histological type | 0.75 | 0.60–0.92 | 0.007 | 0.70 | 0.56–0.87 | 0.002 |
| Cancer status | 7.95 | 3.70–17.09 | 1.08E-07 | 6.93 | 3.20–14.99 | 9.06E-07 |
| Histologic grade | 3.39 | 2.25–5.10 | 4.43E-09 | 2.39 | 1.54–3.69 | 9.62E-05 |
| Gender | 1.03 | 0.72–1.49 | 0.86 | 1.22 | 0.84–1.78 | 0.30 |

**Table 4. Univariate analysis and multivariate analysis of OS prognostic factors in LGG according to CCGA.**

| Parameter | Univariate analysis | | | Multivariate analysis | | |
|---|---|---|---|---|---|---|
| | HR | 95% CI | P | HR | 95% CI | P |
| *PBX3* | 2.43 | 2.06–2.86 | 2.46E-26 | 1.95 | 1.65–2.32 | 2.15E-14 |
| PRS-type | 2.47 | 1.89–3.24 | 5.36E-11 | 1.73 | 1.31–2.28 | 0.0001 |
| Histology | 0.82 | 0.76–0.88 | 3.87E-08 | 0.95 | 0.87–1.04 | 0.272 |
| Grade | 3.17 | 2.38–4.22 | 4.08E-15 | 2.57 | 1.90–3.47 | 6.55E-10 |
| Age | 1.95 | 1.32–2.89 | 0.0008 | 1.43 | 0.95–2.15 | 0.085 |
| IDH-mutation | 0.43 | 0.33–0.57 | 4.76E-09 | 0.80 | 0.59–1.10 | 0.167 |
| 1p19q codeletion | 0.26 | 0.18–0.37 | 3.74E-13 | 0.37 | 0.25–0.55 | 1.29E-06 |
| Gender | 0.96 | 0.73–1.25 | 0.75 | 1.13 | 0.86–1.49 | 0.388 |

**Table 5. Univariate analysis and multivariate analysis of PFI prognostic factors in LGG according to TCGA.**

| Parameter | Univariate analysis | | | Multivariate analysis | | |
|---|---|---|---|---|---|---|
| | HR | 95% CI | P | HR | 95% CI | P |
| *PBX3* | 1.15 | 1.11–1.18 | 3.01E-18 | 1.12 | 1.09–1.16 | 7.94E-12 |
| Age | 2.34 | 1.70–3.21 | 1.48E-07 | 2.30 | 1.67–3.19 | 4.49E-07 |
| Histological type | 0.77 | 0.64–0.91 | 0.002 | 0.74 | 0.62–0.89 | 0.001 |
| Cancer status | 3.52 | 2.35–5.26 | 9.22E-10 | 3.10 | 2.06–4.65 | 5.12E-08 |
| Histologic grade | 1,68 | 1.25–2.26 | 0.0006 | 1,25 | 0,92–1.71 | 0.16 |
| Gender | 0.86 | 0.64–1.15 | 0.31 | 10,94 | 0.70–1.25 | 0.65 |

histologic grade were all high-risk factors (Table 5). And subsequent multivariate analysis shows that *PBX3* (HR = 1.12) remained independently related to PFI, which suggested that *PBX3* might be an independent PFI prognostic factor for LGG patients (Table 5).

## The correlation between *PBX3* expression and the OS of LGG by linear regression model

According to TCGA database, the expression of *PBX3* was positively associated with incident death events in LGG patients. The OR for the death of LGG patients was increased 3.1 times in the high *PBX3* expression group compared with those in the low group (Table 6). Meanwhile, the expression of *PBX3* was negatively correlated with the overall survival time of LGG patients (Table 6). The overall survival time in high group of *PBX3* expression was decreased by 1.1years compared with those in the low group (Table 6).

**Table 6. Association of *PBX3* expression with the OS of LGG according to TCGA.**

| | OS | | OST (year) | |
|---|---|---|---|---|
| | OR (95%CI) | P-value | β (95%CI) | P-value |
| *PBX3* expression | 1.2 (1.1, 1.3) | <0.001 | -0.1 (-0.2, -0.0) | 0.003 |
| *PBX3* tertile | | | | |
| Low | 1 | | 0 | |
| Middle | 1.5 (0.9, 2.4) | 0.143 | -0.1 (-0.6, 0.4) | 0.783 |
| High | 4.1 (2.5, 6.9) | <0.001 | -1.1 (-1.7, -0.5) | <0.001 |

OS: Survival outcome,1: death; OST: Over survival time, unit: year;

**Table 7. Association of *PBX3* expression with the OS of LGG according to CGGA.**

| | OS | | OST (year) | |
|---|---|---|---|---|
| | OR (95%CI) | P-value | β (95%CI) | P-value |
| PBX3 expression | 2.4 (1.9, 3.0) | <0.001 | -1.0 (-1.2, -0.7) | <0.001 |
| PBX3 tertile | | | | |
| Low | 1 | | 0 | |
| Middle | 1.8 (1.2, 2.7) | 0.003 | -0.8 (-1.4, -0.3) | 0.002 |
| High | 4.6 (2.9, 7.2) | <0.001 | -2.0 (-2.6, -1.4) | <0.001 |

OS: Survival outcome,1: death; OST: Over survival time, unit: year.

In addition, we further analyzed the correlation between *PBX3* expression and the OS from CGGA database. the expression of *PBX3* was positively associated with incident death events in LGG patients. The OR for the death events of LGG patients was increased 0.8 and 3.6 times in the middle and high *PBX3* expression group compared with those in the low group (Table 5). Meanwhile, the expression of *PBX3* was negatively correlated with the OS of LGG patients (Table 7). The overall survival time of LGG patients in middle and high group of *PBX3* expression was decreased by 2.0 years compared with those in the low group (Table 7).

## The correlation between *PBX3* expression and the PFI of LGG by linear regression model

According to TCGA database, the expression of *PBX3* was positively associated with disease progression events in LGG patients. The OR for disease progression events 2.6 times in the high *PBX3* expression group compared with those in the low group (Table 8). Meanwhile, the expression of *PBX3* was negatively correlated with the progression free interval time of LGG

**Table 8. Association of *PBX3* expression with the PFI of LGG.**

| | PFI | | PFIT (year) | |
|---|---|---|---|---|
| | OR (95%CI) | P-value | β (95%CI) | P-value |
| *PBX3* expression | 1.2 (1.1, 1.3) | <0.001 | -0.1 (-0.2, -0.1) | <0.001 |
| *PBX3* tertile | | | | |
| Low | 1 | | 0 | |
| Middle | 1.3 (0.8, 2.0) | 0.250 | -0.1 (-0.5, 0.3) | 0.595 |
| High | 3.6 (2.2, 5.7) | <0.001 | -1.0 (-1.5, -0.6) | <0.001 |

PFI: progression free interval outcome,1: disease progression; PFIT: progression free interval time, unit: year.

patients (Table 6). The progression free interval time in high group of *PBX3* expression was decreased by 1 years compared with those in the low group (Table 8).

## GSEA enrichment analysis of *PBX3*

To further analyze the biological functions exerted by *PBX3* in glioma, GSEA enrichment analysis was performed on high and low expression datasets of *PBX3*. The result uncovered significant differences (FDR<0.25, NOM p-value<0.05) in the enrichment of gene sets database (c2. cp.biocarta.v7.5.1.symbols, c2. cp. kegg. v7.5.1 symbols and c5.go.v7.5.1 symbols). Then, the most markedly enriched signaling pathways were screened according to their NES (Fig 6 and Table 9). Fig 6 illustrated that regulation of NOS1 pathway, CK1 pathway, HDAC pathway, Phosphatidylinositol system, Neurotransmitter receptor complex concentrated in *PBX3* high expression phenotype.

## Analysis of gene Co-expression of *PBX3*

Gene correlation analysis by limma package showed that the first five genes with positive correlation with *PBX3* mRNA expression include *GNG12*, *WEE1*, *PDPN*, *FNDC3B* and *TGIF1*,

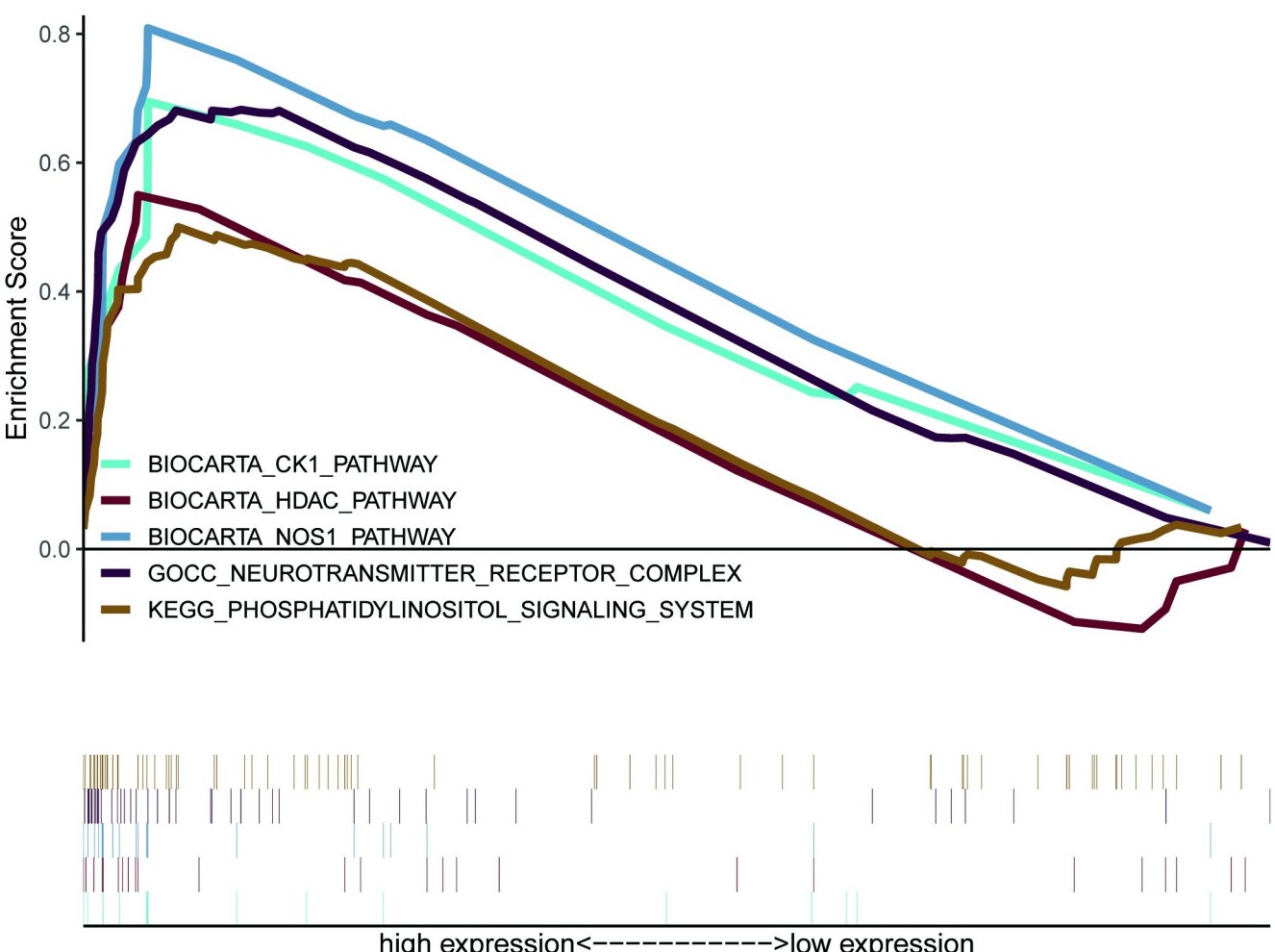

**Fig 6. The integrated GSEA analysis.** Abbreviations: GSEA, gene set enrichment analysis; NES, normalized ES; NOM p-value, normalized p-value; FDR, false discovery rate.

**Table 9. Gene sets enriched in high expression phenotype.**

| Gene set name | ES | NES | NOM p-value | FDR q-value |
|---|---|---|---|---|
| BIOCARTA_NOS1_PATHWAY | 0.81 | 1.81 | 0 | 0.057 |
| BIOCARTA_CK1_PATHWAY | 0.69 | 1.64 | 0.046 | 0.236 |
| BIOCARTA_HDAC_PATHWAY | 0.55 | 1.6 | 0.024 | 0.224 |
| KEGG_PHOSPHATIDYLINOSITOL_SIGNALING_SYSTEM | 0.5 | 1.6 | 0.039 | 0.235 |
| GOCC_NEUROTRANSMITTER_RECEPTOR_COMPLEX | 0.68 | 1.89 | 0.004 | 0.121 |

ES: enrichment score; NES: normalized enrichment score; NOM: nominal; FDR: false discovery rate. Gene sets with NOM p-val<0.05 and FDR q-val<0.25 are considered as significant.

and the first five genes with negative correlation with *PBX3* mRNA expression include *ATP6V1G2*, *AC120036.4*, *PDE2A*, *TEF* and *DNAJC12* (Fig 7).

## Discussion

PBX3 plays an important role in embryonic development, organogenesis and steroid synthesis. There are many similar biological processes in embryonic development and tumorigenesis, such as growth, differentiation and tissues with similar cell groups [13, 14]. Recent studies have shown that *PBX3* is highly expressed in a variety of malignant tumors and is related to tumor cell proliferation, invasion, metastasis and resistance to radiotherapy and chemotherapy [15]. However, the effect of *PBX3* expression on the prognosis of gliomas and the molecular mechanism of its involvement in the occurrence and development of gliomas need to be further studied.

The purpose of this study is to explore the possible prognostic value of *PBX3* in gliomas and the underlying role of *PBX3* in gliomas. Related studies have reported that *PBX3* is highly expressed in glioma tissues [6, 11, 16]. In our study, we found that *PBX3* was highly expressed in glioma tissued by analyzing data from TCGA and GTEx databases. This finding is consistent with the GEPIA database analyses. The result reveals that *PBX3* might acts as a crucial regulatory role in glioma occurrence and progression. The expression of *PBX3* increased with the increase of pathological grade of glioma, this unearthed that *PBX3* might sever as a possible molecular marker for predicting the degree of glioma malignancy. The expression of *PBX3* in recurrent gliomas is significantly higher than that in primary gliomas, indicating that *PBX3* can be used as a tumor marker to predict glioma recurrence. The expression of *PBX3* is related to other clinicopathological features such as idh-mutation,1p19q-codeletion status, age and karnofsky performance score. These results suggest that *PBX3* is associated with tumor progression and is of great value in the diagnosis and prognosis of gliomas.

In addition, we further found that *PBX3* is more valuable in predicting the OS and PFI of LGG patients than that of GBM. Univariate and multivariate Cox regression analysis showed that *PBX3* was a high-risk factor and could be used as an independent prognostic index for patients with LGG glioma. ROC curve analysis indicated that *PBX3* was a predictor of overall survival rate and progression-free survival rate of LGG. the higher the expression of PBX3, the higher the risk of death and disease progression of LGG patients by linear regression model analysis. These results suggest that the expression of PBX3 can be used as a risk indicator to assess death and disease progression of LGG patients.

To further investigate the functions of *PBX3* in glioma, we performed GSEA using TCGA data, GSEA showed that regulation of *NOS1* pathway, *CK1* pathway, *HDAC* pathway, Phosphatidylinositol system, Neurotransmitter receptor complex were differentially concentrated

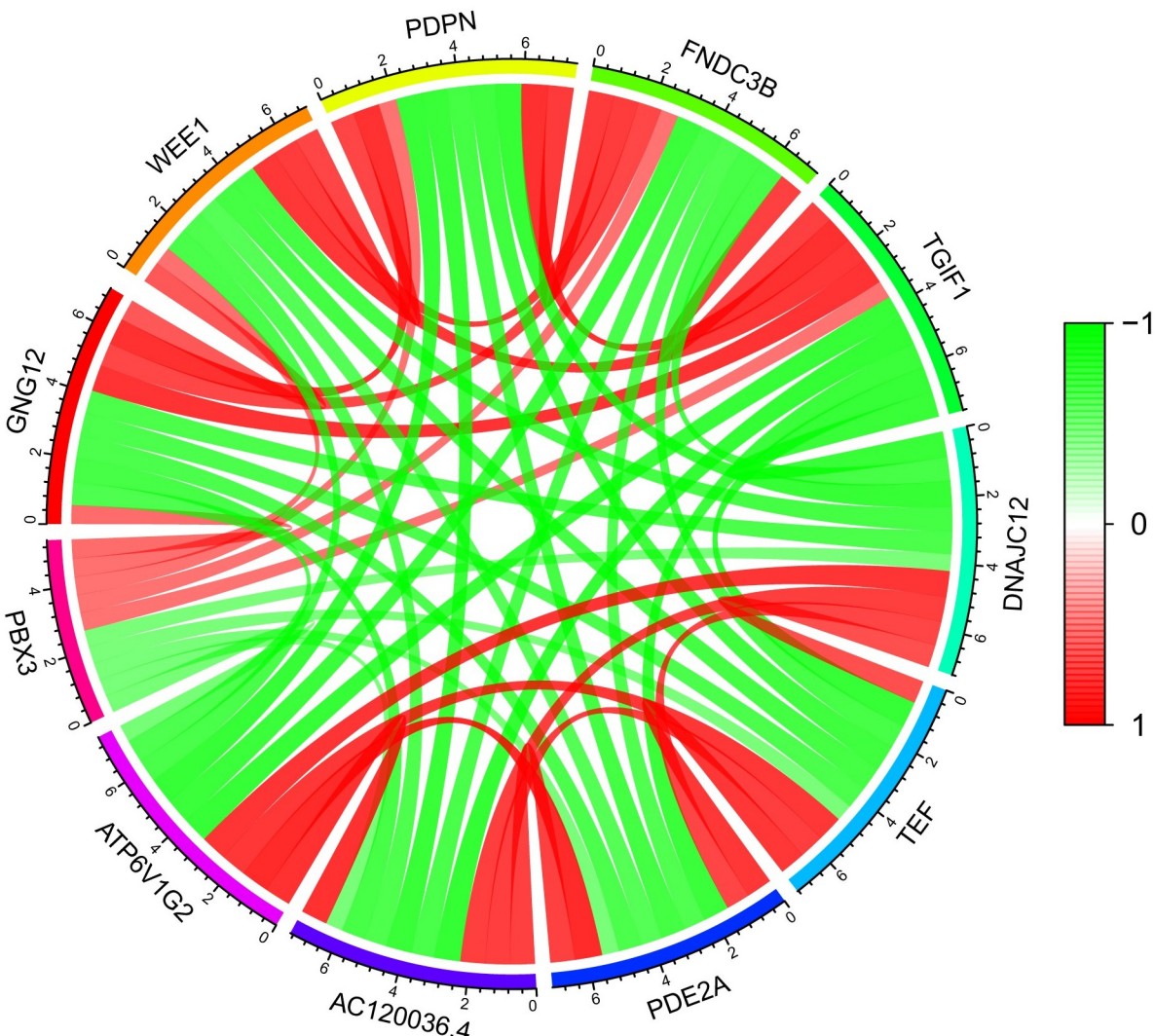

**Fig 7. Analysis of Gene Co-expression of *PBX3* mRNA.** The circle diagram of the genes with positive and negative correlation with *PBX3* mRNA expression (green: negative correlation, red: positive correlation).

in *PBX3* high expression phenotype. *CK1* has been reported involved in the regulation of Wnt pathways, cell proliferation, apoptosis and autophagy [17], dysregulated expression *CK1* iso-forms have previously been linked to tumorigenesis [18]. Histone deacetylases (*HDAC*) deacetylate the histone and induce gene repression thereby leading to cancer [19, 20]. *PI3K* signaling pathway has been as an important signaling pathway involved in the regulation of various aspects of cancers [21–23]. A study by Kotake et al. [24] has reported that low levels of *NO* formed by *NOS1*, triggers cell proliferation primarily via the soluble guanylate cyclase-cyclic guanosine monophosphate (sGC-cGMP) dependent mechanism, and *NOS1* was also observed to be aberrantly expressed in glioma [25]. Diverse neurotransmitters not only maintain normal brain functions but also influence glioma progression [26]. To some extent, these pathways also indicate that *PBX3* is closely related to glioma. The analysis of genetic co-expression of *PBX3* mRNA showed that *PBX3* mRNA was positively correlated with the expression of *GNG12*, *WEE1*, *PDPN*, *FNDC3B* and *TGIF1* genes, and negatively correlated with the

expression of *ATP6V1G2*, *AC120036.4*, *PDE2A*, *TEF* and *DNAJC12* genes. It is reported that *GNG12* [27], *WEE1* [28], *TEF* [29] and *DNAJC12* [30] participate in cell cycle regulation, cell proliferation and cell apoptosis. *PDPN* [31], *FNDC3B* [32], *TGIF1* [33] and *DNAJC12* [30] are involved in cell migration and metastasis. In addition, there are other studies have confirmed that *GNG12* participate in inflammation signals, related to tumor immunity [26]. *PDE2A* is involved in glioma stem cells (GSCs) stemness and tumorigenesis [34]. Therefore, we speculate that the high expression of *PBX3* in glioma cells may be participated in a variety of biological processes, including cell proliferation, cell apoptosis, cell migration and metastasis, tumor immunity and stem cells stemness through the above signal pathways and co-expressed molecules.

Our study demonstrated that the expression level of *PBX3* is related to the prognosis of glioma patients, and the high expression of *PBX3* is associated with poor prognosis and is a high-risk factor. Compared with GBM patients, *PBX3* is more valuable in predicting the survival time of LGG patients. *PBX3* can be used as an independent prognostic index for patients with LGG glioma. We further explored the signal pathways that *PBX3* may be involved in and the genes co-expressed with *PBX3*. The study explored the relationship between *PBX3* expression and glioma via bioinformatics, and needed further study with function and mechanism of *PBX3* by a range of vivo and vitro experiments.

## Supporting information

**S1 Fig. Expression level of *PBX3* in normal brain, glioma tissues and glioma cells by analyzing data from TCGA, GTEx, GEPIA and CCLE databases.** (A-C) Expression level of *PBX3* in full-grade gliomas, LGG and GBM in comparison with the normal brain tissues according to TCGA and GTEx. (D) Expression level of *PBX3* in LGG and GBM in comparison with the normal brain tissues according to GEPIA. (E) and (F) The expression level of *PBX3* between LGG and GBM according to TCGA and CGGA databases. (G) and (H) The expression level of *PBX3* in histological types of gliomas according to TCGA and CGGA databases. (I) The expression level of *PBX3* in glioma cell lines according to CCLE databases.
(ZIP)

**S2 Fig. Correlation analysis between PBX3 differential expression and clinical features according to TCGA and CGGA.** A, histologic-grade by TCGA; B, histologic-grade by CGGA; C, sample type by TCGA; D, sample type by CGGA; E, IDH-mutant by TCGA; F, IDH-mutant by CGGA; G, Age by TCGA; H, Age by CGGA; (I and J), cancer-status and karnofsky performance score by TCGA; K, 1p19q-codeletion by CGGA.
(ZIP)

**S3 Fig. Relationship between *PBX3* expression and OS of patients with glioma.** (A and B), full-grade gliomas according to TCGA and CGGA databases;(C-E), LGG according to TCGA, CGGA and GEPIA databases;(F-H), GBM according to TCGA, CGGA and GEPIA databases.
(ZIP)

**S4 Fig. Relationship between *PBX3* expression and PFI of patients with glioma.** (A), full-grade gliomas according to TCGA databases;(B and C), LGG according to TCGA and GEPIA databases;(D and E), GBM according to TCGA and GEPIA databases.
(ZIP)

**S5 Fig. ROC curve analysis of the predictive efficacy of *PBX3* mRNA on 1 -, 3-and 5-year overall survival rate and progression-free survival rate of LGG.** (A)and(B), overall survival

rate;(C) progression-free survival rate.
(ZIP)

**S6 Fig. The integrated GSEA analysis.** Abbreviations: GSEA, gene set enrichment analysis; NES, normalized ES; NOM p-value, normalized p-value; FDR, false discovery rate.
(ZIP)

**S7 Fig. Analysis of Gene Co-expression of *PBX3* mRNA.** The circle diagram of the genes with positive and negative correlation with *PBX3* mRNA expression (green: negative correlation, red: positive correlation).
(XLS)

**S1 Table.**
(CSV)

**S2 Table.**
(CSV)

**S3 Table.**
(CSV)

**S4 Table.**
(XLSX)

**S5 Table.**
(XLSX)

**S6 Table.**
(XLSX)

**S7 Table.**
(ZIP)

## Acknowledgments

We would like to thank the peer reviewers for their constructive and insightful comments.

## Author Contributions

**Conceptualization:** Cuicui pan, Yuanquan Si, Yueran Zhao.

**Data curation:** Cuicui pan, Na Li, Ni Zheng.

**Formal analysis:** Cuicui pan, Xueli bai, Na Li.

**Funding acquisition:** Cuicui pan.

**Investigation:** Cuicui pan, Xueli bai.

**Methodology:** Cuicui pan, Xueli bai, Na Li, Ni Zheng, Yueran Zhao.

**Project administration:** Cuicui pan.

**Resources:** Cuicui pan, Xueli bai, Na Li, Ni Zheng.

**Software:** Cuicui pan, Na Li.

**Supervision:** Cuicui pan, Ni Zheng, Yuanquan Si.

**Validation:** Cuicui pan, Yuanquan Si, Yueran Zhao.

**Visualization:** Cuicui pan, Yuanquan Si, Yueran Zhao.

**Writing – original draft:** Cuicui pan.

**Writing – review & editing:** Cuicui pan, Yuanquan Si, Yueran Zhao.

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
