## [Decision Letter · Decision Letter 0]

16 Jan 2023

PONE-D-22-30990PBX3 as a biomarker for the early diagnosis and prediction of prognosis of gliomaPLOS ONE

Dear Dr. pan,

Thank you for submitting your manuscript to PLOS ONE. After careful consideration, we feel that it has merit but does not fully meet PLOS ONE’s publication criteria as it currently stands. Therefore, we invite you to submit a revised version of the manuscript that addresses the points raised during the review process.

We look forward to receiving your revised manuscript.

Kind regards,

Amit Kumar Pandey, Ph.D.

Academic Editor

PLOS ONE

Journal Requirements:

Reviewers' comments:

Reviewer's Responses to Questions

**Comments to the Author**

1. Is the manuscript technically sound, and do the data support the conclusions?

Reviewer #1: Partly

Reviewer #2: Yes

2. Has the statistical analysis been performed appropriately and rigorously? 

Reviewer #1: N/A

Reviewer #2: Yes

3. Have the authors made all data underlying the findings in their manuscript fully available?

Reviewer #1: Yes

Reviewer #2: Yes

4. Is the manuscript presented in an intelligible fashion and written in standard English?

Reviewer #1: Yes

Reviewer #2: Yes

5. Review Comments to the Author

Reviewer #1: Identification of new biomarkers of glioma and glioblastoma is important scientific problem. The authors focused on transcription factor PBX3 which was shown to increase the aggressiveness of glial tumors in in vitro tests. The problem is that the authors present only analysis of public databases and do not provide any experimental data supporting their bioinformatic analysis. The thing is that TCGA combines data from multiple datasets which should be carefully evaluated prior to inclusion to analysis. The information concerning treatment of glioma and glioblastoma patients, age and gender distribution grade/size of tumor and other parameters is pretty much absent in the manuscript. Another problem is that PBX3 likely is a global regulator of Pol2-driven transcription similar to more studied proteins of PBX family. Therefore, it is easy to guess that its upregulation is rather a consequence of malignant transformation, but not a reason for faster growth of tumors. It would be more interesting to include information concerning correlation of PBX3 expression and treatment efficiency as well as relapse time since almost each glioblastoma patient suffers from fast relapse.

Lack of experimental confirmation of bioinformatic analysis on smaller but better defined cohorts, primary glioma cell lines with different phenotypes or glioma xenotransplantation models greatly complicates the evaluation of presented data.

Based on that, I would recommend to send this manuscript to more specialized bioinformatics, biomarker or clinical data analysis journal, since the manuscript in its current form does not fit to the scope of the PLOS One. My concern is that the manuscript will not attract sufficient reader's interest, and I can not support its acceptance to PLOS One.

Reviewer #2: The findings are interesting and technically well performed. Specific points that the authors need to address are as follows:

1. The mechanisms(s) that can lead to enhanced expression of PBX3 in glioma should be investigated.

2. The functional significance of PBX3 overexpression should be validated by knockout experiments using siRNa/shRNA.

3. A limited in vivo study will greatly increase the impact of the findings.

4. The authors should provide their own justification and relevance of the study. This will help the readers to understand the importance of the paper.

5. Typographical errors were found throughout the manuscript and should be corrected.

6. PLOS authors have the option to publish the peer review history of their article (what does this mean?). If published, this will include your full peer review and any attached files.

Reviewer #1: No

Reviewer #2: No

---

## [Author Response · Author response to Decision Letter 0]

25 Mar 2023

Response to Reviewer #1: The main purpose of this study is to explore PBX3 may be involved in the occurrence and development of glioma, and has potential reference value for the early diagnosis and prediction of prognosis of glioma. First, in this study, the high expression of PBX3 in glioma was confirmed by analyzing the data of GTEX and TCGA database, and the above conclusion was further confirmed by GEPIA online database. Second. PBX3 was observably related with disease type, histological types, histologic-grade, cancer status, IDH mutation, age, karnofsky performance score and 1p19q-codel according to TCGA and CGGA two major databases. And, by analyzing the expression level of PBX3 in 57 glioma cell lines from CCLE database, it was found that the expression of PBX3 was increased in glioma cell lines. The above databases are recognized by the world, and the data are reliable, so the conclusions drawn by analyzing the data of these databases are very reliable. Third, multi-statistical methods such as the univariable survival and multivariate Cox analysis, ROC curve survival analysis, and Linear regression analysis by R and EmpowerStats was used to confirm the conclusion of the value for the early diagnosis and prediction of prognosis of glioma. A variety of statistical methods confirm each other, and the conclusions are consistent, so the conclusion of this study is reliable. In addition, TCGA data were downloaded for GSEA and Co-expression analyses, which was performed to study the function of PBX3. This study provided valuable data for the follow-up function research of PBX3. Last, the information concerning treatment of glioma and glioblastoma patients, age and gender distribution grade of tumor and other parameters has been presented in the manuscript by table. The experimental data supporting their bioinformatic analysis and correlation of PBX3 expression and treatment efficiency as well as relapse time will be done in our follow-up experiments. Therefore, this study is very valuable and will arouse enough interest from readers.

Response to Reviewer #1:

Thank you very much for giving such valuable advice to this research and pointing out the research direction for my subsequent experiments. The mechanisms that can lead to enhanced expression of PBX3 in glioma，the functional significance of PBX3 overexpression and in vivo study will be will be continued in follow-up experiments. I have corrected the typographical error in the manuscript. In our study, on the first, we found that PBX3 was highly expressed in glioma tissue, the expression of PBX3 increased with the increase of pathological grade of glioma, the expression of PBX3 in recurrent gliomas is significantly higher than that in primary gliomas, and the expression of PBX3 is related to other clinicopathological features such as idh-mutation,1p19q-codeletion status, age and karnofsky performance score. These results suggest that PBX3 is associated with tumor progression and is of great value in the diagnosis and prognosis of gliomas. Second, we used univariate and multivariate Cox regression analysis, ROC curve analysis and linear regression model analysis further confirmed that the expression of PBX3 can be used as a risk indicator to assess death and disease progression of glioma. At last, we further investigate the functions of PBX3 in glioma by GSEA and co-expression analysis. The research is progressive and interrelated, and is easy to understand with the readers.

---

## [Decision Letter · Decision Letter 1]

26 Sep 2023

PONE-D-22-30990R1PBX3 as a biomarker for the early diagnosis and prediction of prognosis of gliomaPLOS ONE

Dear Dr. Si,

Thank you for submitting your manuscript to PLOS ONE. After careful consideration, we feel that it has merit but does not fully meet PLOS ONE’s publication criteria as it currently stands. Therefore, we invite you to submit a revised version of the manuscript that addresses the points raised during the review process.

Based on the one reviewer's comments, please submit your revised manuscript by Nov 10 2023 11:59PM. If you will need more time than this to complete your revisions, please reply to this message or contact the journal office at plosone@plos.org. Please include the following items when submitting your revised manuscript:A rebuttal letter that responds to each point raised by the academic editor and reviewer(s). You should upload this letter as a separate file labeled 'Response to Reviewers'.A marked-up copy of your manuscript that highlights changes made to the original version. You should upload this as a separate file labeled 'Revised Manuscript with Track Changes'.An unmarked version of your revised paper without tracked changes. You should upload this as a separate file labeled 'Manuscript'.If applicable, we recommend that you deposit your laboratory protocols in protocols.io to enhance the reproducibility of your results. Protocols.io assigns your protocol its own identifier (DOI) so that it can be cited independently in the future. For instructions see: https://journals.plos.org/plosone/s/submission-guidelines#loc-laboratory-protocols. Additionally, PLOS ONE offers an option for publishing peer-reviewed Lab Protocol articles, which describe protocols hosted on protocols.io. Read more information on sharing protocols at https://plos.org/protocols?utm_medium=editorial-email&utm_source=authorletters&utm_campaign=protocols.

We look forward to receiving your revised manuscript.

Kind regards,

Girijesh Kumar Patel, PhD

Academic Editor

PLOS ONE

Journal Requirements:

Reviewers' comments:

Reviewer's Responses to Questions

**Comments to the Author**

1. If the authors have adequately addressed your comments raised in a previous round of review and you feel that this manuscript is now acceptable for publication, you may indicate that here to bypass the “Comments to the Author” section, enter your conflict of interest statement in the “Confidential to Editor” section, and submit your "Accept" recommendation.

Reviewer #2: All comments have been addressed

Reviewer #3: All comments have been addressed

2. Is the manuscript technically sound, and do the data support the conclusions?

Reviewer #2: Yes

Reviewer #3: Yes

3. Has the statistical analysis been performed appropriately and rigorously? 

Reviewer #2: Yes

Reviewer #3: Yes

4. Have the authors made all data underlying the findings in their manuscript fully available?

Reviewer #2: Yes

Reviewer #3: Yes

5. Is the manuscript presented in an intelligible fashion and written in standard English?

Reviewer #2: Yes

Reviewer #3: Yes

6. Review Comments to the Author

Reviewer #2: The authors have extensively revised the manuscript and it can be accepted now. All the queries raised have been addressed.

Reviewer #3: In the current manuscript, Pan et al. studied PBX3 and proposed it as a biomarker for the early diagnosis and prediction of prognosis for the patients with Glioma. The study is designed and executed well. However, there are some minor concerns need to be addressed-

1. Abbreviation should be used at first use and followed thereafter throughout the manuscript. E.g. LGG, GBM, TCGA, GEPIA, CGGA, CCLE, etc,. Therefore, it is recommended to provide a list of abbreviations.

2. Authors proposed PBX3 as a diagnostic marker, However, it is a transcription factor and may not be present in the biological fluids freely. Therefore, It is important to discuss how PBX3 can be used as a diagnostic marker?

7. PLOS authors have the option to publish the peer review history of their article (what does this mean?). If published, this will include your full peer review and any attached files.

Reviewer #2: No

Reviewer #3: No

---

## [Author Response · Author response to Decision Letter 1]

10 Oct 2023

1 Abbreviation should be used at first use and followed thereafter throughout the manuscript. E.g. LGG, GBM, TCGA, GEPIA, CGGA, CCLE, etc,. Therefore, it is recommended to provide a list of abbreviations.

Response:Thank you for your feedback and suggestions. We appreciate your guidance regarding the inclusion of an abbreviation list. We have taken your suggestion into consideration, and we have added an abbreviation list at the end of the main text of the manuscript.We believe that this addition will enhance the readability and understanding of the manuscript for the readers. Thank you once again for bringing this to our attention, and we hope that this revision meets your expectations.

2 Authors proposed PBX3 as a diagnostic marker, However, it is a transcription factor and may not be present in the biological fluids freely. Therefore, It is important to discuss how PBX3 can be used as a diagnostic marker?

Response: Thank you for your valuable comments and suggestions. We appreciate your insight regarding the proposed diagnostic marker PBX3.Indeed, PBX3 is known to be a transcription factor, primarily involved in regulating gene expression. We acknowledge that the expression of PBX3 in bodily fluids is not easily detectable. However, existing literature has reported on the regulatory mechanisms of PBX3 expression, which provide evidence for its diagnostic value. Such as, Hongru Sun[1] et al confirmed that PBX3 hypermethylation in peripheral blood leukocytes predicts better prognosis in colorectal cancer. In addition,we will further explore the possibility of indirectly assessing PBX3 activity or downstream effects in biological samples, such as measuring its target genes or associated signaling pathways. Additionally, we will emphasize the importance of considering alternative methods or strategies for evaluating PBX3's diagnostic utility, including tissue-based assessments or other complementary biomarkers.We thank you again for bringing up this important point, and we will revise our manuscript accordingly to provide a more comprehensive discussion on how PBX3 can be utilized as a diagnostic marker

References

1 Hongru Sun , Hao Huang , Dapeng Li , Lei Zhang , Yuanyuan Zhang, Jing Xu , Ying Liu , Yupeng Liu , Yashuang Zhao. PBX3 hypermethylation in peripheral blood leukocytes predicts better prognosis in colorectal cancer: A propensity score analysis. Cancer Med.2019 Jul;8(8):4001-11.

---

## [Editor Report · Decision Letter 2]

18 Oct 2023

PBX3 as a biomarker for the early diagnosis and prediction of prognosis of glioma

PONE-D-22-30990R2

Dear Dr. Si,

We’re pleased to inform you that your manuscript has been judged scientifically suitable for publication and will be formally accepted for publication once it meets all outstanding technical requirements.

Kind regards,

Girijesh Kumar Patel, PhD

Academic Editor

PLOS ONE
---

## [Editor Report · Acceptance letter]

14 Nov 2023

PONE-D-22-30990R2 

*PBX3* as a biomarker for the early diagnosis and prediction of prognosis of glioma 

Dear Dr. Si:

I'm pleased to inform you that your manuscript has been deemed suitable for publication in PLOS ONE. Congratulations! Your manuscript is now with our production department. 

Kind regards, 

on behalf of

Dr. Girijesh Kumar Patel 

Academic Editor

PLOS ONE